# Aggressive Angiomyxoma of the Lower Female Genital Tract: A Review of the MITO Rare Tumors Group

**DOI:** 10.3390/cancers16071375

**Published:** 2024-03-31

**Authors:** Miriam Dellino, Francescapaola Magazzino, Lavinia Domenici, Stefania Cicogna, Salvatora Tindara Miano, Sandro Pignata, Giorgia Mangili, Gennaro Cormio

**Affiliations:** 1Department of Interdisciplinary Medicine (DIM), University of Bari “Aldo Moro”, Piazza Giulio Cesare 11, 70124 Bari, Italy; gennaro.cormio@uniba.it; 2Complex Operating Unit Ginecologia E Ostetricia, Ospedale Civile Di San Dona’ Di Piave (Venezia), Aulss4 Veneto Orientale, 30027 San Donà di Piave, Italy; francescapaola.magazzino@aulss4.veneto.it; 3Division of Obstetrics and Gynecology, Azienda Ospedaliera Universitaria Pisana, University of Pisa, 56126 Pisa, Italy; lavinia.domenici@gmail.com; 4Institute for Maternal and Child Health-IRCCS “Burlo Garofolo”, 34145 Trieste, Italy; stefania.cicogna@asltrieste.it; 5Complex Operating Unit of Oncology, Azienda Ospedaliera Universitaria Senese, 53100 Siena, Italy; s.miano@ao-siena.toscana.it; 6Department of Urology and Gynecology, Istituto Nazionale Tumori IRCCS ‘Fondazione G Pascale’, 80144 Napoli, Italy; s.pignata@istitutotumori.na.it; 7Obstetrics and Gynecology Unit, IRCCS San Raffaele Scientific Institute, 20132 Milan, Italy; mangili.giorgia@hsr.it; 8Gynecologic Oncology, IRCCS Istituto Tumori “Giovanni Paolo II”, 70124 Bari, Italy

**Keywords:** aggressive angiomyxoma, deep angiomyxoma, vulva

## Abstract

**Simple Summary:**

Aggressive angiomyxoma is a mesenchymal tumor with localized aggressiveness, affecting the connective tissue of the perineum or the lower pelvis. Prevalence in the population is unknown due to its rarity, making management and counseling difficult. The management of angiomyxoma includes multiple types of treatment, such as radical surgery with tumor-free margins, but the probability of local recurrence is high, despite extensive excision with unscathed mar-gins. Considering the low mitotic activity of angiomyxoma, there is not always a rationale for adjuvant radiotherapy and chemotherapy. Given its exceptionally low incidence, optimal management of the disease remains a subject of on-going debate, and a unanimous consensus on treatment strategies has yet to be reached.

**Abstract:**

Aggressive angiomyxoma (AAM) is a rare, locally aggressive, myxoid mesenchymal neoplasm primarily found in the pelvic and perineal regions of young adult females. It is a slow growing and locally infiltrating tumor. Preoperative diagnosis is difficult due to the rarity of these tumors and absence of characteristic signs and symptoms. The primary management is tumor excision. Incomplete excision is common because of the infiltrating nature of the neoplasm and absence of a definite capsule. Other non- surgical modalities have been employed, such as radiotherapy, embolization, GnRH analogues or other anti-estrogenic agents. Local relapses occur in 30–40% of the cases, and often appear many years (sometimes decades) after the first excision. Occasional distant metastasis has also been reported. A limited number of cases have been reported in the literature, mostly in the form of small case series or isolated case reports. Therefore, the aim of this paper by a team of experts from the MITO rare tumors group is to review clinical findings, pathologic characteristics and outcome of patients affected by this rare condition in order to be able to offer up-to-date guidance on the management of these cases.

## 1. Introduction

Aggressive angiomyxoma (AAM) is a mesenchymal tumor with localized aggressiveness, affecting the connective tissue of the perineum or the lower pelvis [1]. The primary occurrence is among women of reproductive age, with a female-to-male ratio of 6.6/1 [2]. The term “aggressive” is used to highlight its potential infiltrative behavior. Despite its name, AAM does not have an aggressive nature, since its capacity to metastasize and the histologic features of this tumor do not suggest a more aggressive pattern than the commonest vulva carcinoma with also a considerably better prognosis [3,4,5]. Indeed, the pelvis, perineum, vulva, vagina, and bladder are the most common anatomic sites involved [2]. On the other hand, it is important to underline that AAM exhibits an angioinvasive behavior, but with a slow growth and a high rate of local recurrence, and a limited ability to metastasize. The origin is myofibroblastic differentiation of spindle or stellate cells divided by myxoid stroma and abundant vascular constituents. The management of angiomyxoma includes multiple types of treatment, such as radical surgery with tumor-free margins, but the probability of local recurrence is high, despite extensive excision with unscathed margins [6]. Given its exceptionally low incidence, optimal management of the disease remains a subject of ongoing debate, and a unanimous consensus on treatment strategies has yet to be reached [7]. Consequently, we present here a systematic review that may serve as a valuable resource for further discussion and future clinical practice guidelines.

## 2. Materials and Methods

We conducted a systematic review of DAM case reports by searching electronic databases including PubMed, the Cochrane Library, Embase, Web of Science, and Medline. The article research adhered to the Preferred Reporting Items for Systematic Reviews and Meta-Analyses guidelines (Figure 1) [8]. The following search terms were used: “aggressive angiomyxoma”, “deep angiomyxoma”, “vulvar”. No limitations were imposed on the publication timeframe. Specifically, we considered case series and case reports published in English. Three authors (M.D., F.P.M., S.C.) independently reviewed the titles and abstracts of eligible articles, eliminating duplicates. The full texts of potentially suitable studies were then independently evaluated for eligibility by two authors. Any discrepancies were resolved through discussion with two senior reviewers (G.C. and G.M.). Data were collected from articles published between 1983 (when Steeper and Rosai first described a case of aggressive angiomyxoma of the vulva) and February 2023. Articles reporting DAM in pregnant women were excluded.

## 3. Pathological Examination

From a macroscopic perspective, these tumors frequently exhibit a smooth surface which is partially or entirely encapsulated. The cut surface presents a shiny, gelatinous appearance with a bluish grey hue, often accompanied by regions of hemorrhage and congestion [9]. Size is variable, although usually the maximum diameter is 10–20 cm. They may produce pressure on adjacent organs. They are usually homogenous in consistency with no obvious nodularity [9] (Figure 2).

Microscopically, the tumor exhibits spindle and stellate-shaped cells within a myxoid matrix characterized by delicate wavy collagen fibrils [10]. Additionally, there is a notable presence of vascular structures of various sizes [10]. The cells have abundant wispy pink cytoplasm with bland nuclei [10]. There is no cytological atypia, no atypical mitotic features or discernible mitotic activity, nor any evidence of coagulative tumor cell necrosis [11]. Immunohistochemical analyses typically reveal positive reactivity for desmin, smooth muscle actin (SMA), muscle specific actin, vimentin, CD34, estrogen, and progestin receptors in the majority of these tumors [11]. Conversely, the S100 protein is invariably negative. The vast majority of these neoplasms exhibit positivity for estrogen and progesterone receptors, indicating that AA is likely a hormone-dependent tumor. This is supported by observations of rapid growth and recurrence during pregnancy [12]. Recurrent tumors usually show similar histological characteristics. Diagnostic problems may arise when the pathologist is dealing with uncommon morphological features [13], deposition with or without hyalinized blood vessels, or neurofibroma-like appearance [12].

## 4. Clinical Features

The clinical aspects described in the world literature are of a bulky and soft tumor which looks like a cutaneous mass, or of an ulcerated or polypoid/pedunculated tumor, sometimes associated with bleeding [14]. The areas most affected are the labia majora, vulva and pubis, perineum, medial gluteal region, and the periurethral region [15,16]. Their size ranges from 1 to 60 cm, but the chances of recurrence do not depend on the size [15]. Almost all patients report a soft mass on the pelvic region, sometimes bilobated, mobile, with slow growth, pelvic fullness, perineal bulge, and discomfort/dyspareunia [14]. The clinical presentation is similar to other benign lesions such as Bartholin duct cysts, lipoma, vulvar masses or abscesses, or Gartner duct cysts. Consequently, AAM can often be misdiagnosed [15].

## 5. Management: Treatment of Aggressive Angiomyxoma of the Vulva

Several therapy modalities have been described for the treatment of angiomyxoma. Radical surgery with tumor-free margins is the treatment of choice [17]. The literature has highlighted how the probability of local recurrence is very high even with wide local excision. On the other hand, we see how adjuvant radiotherapy and chemotherapy have limited indications considering the low mitotic activity of angiomyxoma [18]. Since most of these tumors have positive estrogen and progesterone receptors, they may respond to hormone treatment with gonadotropin-releasing hormone (GnRH), raloxifene, and tamoxifen agonists, both preoperatively and after relapse [14]. Therefore, several studies have published the use of GnRH agonists as the sole treatment or as adjuvant hormone therapy after surgery for the management of these malignancies. However, the duration is not clearly defined GnRH agonist or antihormone therapy (tamoxifen) for correlation with hormone-based proliferation are indicated as emerging therapies [8]. This treatment is indicated either as an adjuvant approach for residual masses or before surgery to minimize the size of the tumor and promote the possibility of complete excision [15]. However, after approximately 10 days of treatment, receptors undergo down-regulation through internalization, leading to decreased hormone levels [8]. Indeed, some reports have documented complete radiological resolution of tumors with the administration of GnRH agonists, both in cases of primary and recurrent tumors. For this reason, some authors advocate for the inclusion of GnRH agonists in the treatment protocol to potentially avoid the necessity for radical pelvic surgery in hormone receptor-positive patients [18]. More recently, evidence has emerged to support the use of an aromatase inhibitor in angiomyxoma treatment; a study reported successful tumor shrinkage when the inhibitor was administered prior to resection [17]. This treatment approach works by blocking the aromatase enzyme from synthesizing estrogen. Clearly, adjuvant therapy in raloxifene, tamoxifen, or GnRH agonists like leuprolide acetate and goserelin have demonstrated efficacy in cases where the tumor exhibits sensitivity to estrogen and progesterone receptors [8]. It must be considered, however, that these drugs have multiple side effects including bone depletion and menopausal-like manifestations [17]. It is also highlighted that following the discontinuation of the drug, there is the subsequent growth of the tumor [14], therefore, these drugs do not represent a definitive treatment [15]. An alternative option in postmenopausal women is to use oral hormone therapy based on aromatase inhibitors [17]. Finally, angiographic embolization or chemoembolization has been described [14]. Extensive surgical excision with tumor-free margins remains the gold standard. It is necessary to underline the need for long-term follow-up due to the high recurrence frequency of between 36 and 72% which can be found at the same site as the initial resection [15]. Therefore, the widest margin of excision as possible is recommended without undue morbidity [17]. In fact, growth is usually slow and locally infiltrative, extending insidiously into adjacent soft tissues [14]. The presence of an infiltrative capacity of this tumor and the absence of a well-defined capsule limits the incomplete removal of the neoplasm and contributes to the high rate of tumor recurrence [18]. Distant metastases are very rare thanks to the low mitotic index, and so the prognosis is excellent [17]. Radiological investigations are fundamental for diagnosis, monitoring of recurrences, and planning of surgery in order to examine the actual extent of the tumor [17]. Treatment alternatives are angiographic embolization, mainly as adjuvant therapy or in cases where surgery is contraindicated [15]. The role of chemotherapy and radiotherapy has limited indications [15]. For example, the use of radiotherapy can be proposed in cases of multiple recurrent diseases after poor results with surgical excision [15,17,18]. Indeed, to date there is no unequivocal consensus in the literature on the appropriate treatment method. Therefore, treatment should be tailored on a case-by-case basis after discussion in a multidisciplinary and experienced team.

## 6. Prognostic Factors

AAM usually exhibit a slow, insidious growth pattern, a capacity for local infiltration, and a marked tendency for repeated local recurrence [8,19]. The reported local recurrence rate stands at around 30% and may occur months to several years after excision (2 months to 15 years) [18]. Multiple recurrences are not infrequent 3–6, and more than half of the presenting lesions and all relapsing lesions involve adjacent organs, with 71% of recurrences occurring within three years of resection [18].

Despite a quite high relapse rate, distant metastasis and death from disease are extremely rare [18]. Two cases of distant metastasis have been reported in the literature [18]:-a case of an aggressive angiomyxoma of the pelvis, with massive bilateral pulmonary, mediastinal, iliac, and aortic lymph node and peritoneal metastases ending in death described by Siassi et al. in 1999 [14].-another case, a 34-year-old woman developed several local recurrences after primary resection of an AAM and subsequently died from multiple lung metastases [17].

Notwithstanding its capacity to metastasize, the histologic features of this tumor do not suggest a more aggressive pattern and prognosis is considerably good [15]. Most relapses were related to incomplete resection. However, these tumors often invade surrounding tissues and the visceral peritoneum, making complete resection quite challenging [18]. Except for positive surgical margins, there are no clinical or histological predictors for tumor recurrence. Nonetheless, Chan et al. found that patients with negative resection margins have similar chances of remaining disease-free compared with those having positive resection margins, 50% vs. 40% in 10 years [8]. Moreover, a review of over 100 cases observed that those with positive margins are as likely to have recurrences as those with negative margins [18]. Recently, several studies have assessed that incomplete or partial resection could be acceptable, especially when high operative morbidity is anticipated and preservation of fertility represents an issue [20]. In a recent study by Li et al., outcomes of 14 females with AA were analyzed during a median follow-up of 78.8 months [21]. Univariate Cox regression analysis identified tumor margin (*p* = 0.012) and initial treatment site (*p* = 0.039) as being associated with disease-free survival (DFS) [21]. Patients with positive tumor margins had a significantly lower probability of survival with DFS than those with negative margins (HR = 3.41, CI: 2.73–15.74, *p* = 0.012). The authors hypothesized that tumor location, tumor margin, surgical procedure, and tumor size had a greater effect on patient outcomes (Figure 3).

In particular, the prognosis of patients varied depending on the tumor location. For instance, patients with pelvic tumors had the worst prognosis, while patients with vaginal tumors had a better prognosis [22]. In addition, Li and colleagues observed that patients with tumor sizes in the range of 5–10 cm had the poorest probability of DFS survival, while those with tumor size less than 5 cm had the greatest probability of DFS survival. Risk of relapse was demonstrated to be higher among patients older than 34 years (*p* = 0.67) [21].

## 7. Results

From the literature search, we identified 17 articles reporting a total of 19 cases listed in Table 1.

## 8. Discussion

Soft tissue myxoid tumors constitute a diverse array of lesions characterized by varying degrees of extracellular myxoid matrix [22]. Steeper and Rosai were the first to describe nine cases of a distinct, infiltrative, locally aggressive but non-metastasizing fibro-myxoid soft tissue tumors arising in the pelvic and perineal regions of young female patients, which they designated aggressive angiomyxoma [1]. This category of tumors displays considerable variability in their biological behavior and includes benign lesions, tumors prone to recurrence but lacking metastatic potential, and outright malignant neoplasms [31]. In general, angiomyxomas are classified as either superficial (also referred to as cutaneous myxoma) or AA. Superficial angiomyxoma is commonly associated with the Carney complex [46]. This type of lesion predominantly affects middle-aged adults and can manifest in superficial tissues throughout the body, but mostly it occurs in the trunk, lower extremities, and head and neck regions [23]. Clinically, most lesions emerge as slow growing polypoid cutaneous lesions and often have to be differentiated from angiomyoblastoma, myxoid neurofibroma, myxoma, spindle cell lipoma, myxoid liposarcoma, leiomyosarcoma, and botryoid rhabdo-myosarcoma [24]. Aggressive angiomyxoma typically displace neighboring organs without direct invasion, yet their locally infiltrative nature can eventually result in the invasion of adjacent organs over time, culminating in the development of a large tumor that occupies the abdominal and pelvic cavities [18]. The recurrence rate has been reported to be as high as 70%; with the majority of cases recurring within two years. However, recurrence can occur as early as a few months or as late as 20 years after initial treatment [22,26]. Evidence suggests that AAM affects almost exclusively the genital, perineal, and pelvic regions in women of reproductive age, implicating especially the vulva [27]; less frequently, the buttocks, retroperioneum, and inguinal area may be involved. AAM are infrequently observed in men, with the scrotum being the primary site of involvement [28]. The prevailing theory regarding AAM pathogenesis suggests that the lesion originates from a primitive multipotent mesenchymal cell found in the lower female genital tract, with the capacity for diverse differentiation pathways. [20].

Interesting molecular studies have detected a significant clonal aberration of chromosome 12, in the region 12q13–15, with rearrangement of the HMGIC gene (high-mobility group protein isoform I-C) linked with AAM [29]. Thus, AAM is molecularly part of the benign group of mesenchymal tumors showing multiple aberration region involvement [30]. Immunohistochemically, AA cells show diffuse staining for estrogen receptor (ER), progesterone receptor (PR), desmin, smooth muscle actin, and vimentin. Numerous tumors are also CD34-positive [31]. Furthermore, AAM seems to grow in size in pregnant women due to the positivity of progesterone receptors, underlying a concrete hormonal correlation between the tumor and female hormonal status [20]. Hormonal correlation suggests that antihormonal therapy (e.g., Tamoxifen), gonadotrophin-releasing hormone (GnRH) agonist, or aromatase inhibitors could be considered feasible emerging options in AA treatment [28]. Moreover, a neo-adjuvant treatment to reduce tumor size before surgery, facilitating complete excision, or an adjuvant approach for incompletely resectable/residual mass could be investigated [29]. However, the adverse effects of long-term use of the GnRH agonist (e.g., menopausal symptoms and bone loss) and tumor regrowth after drug interruption do not allow us to approve it as a best choice of treatment [32,33,45]. Large surgical excision with tumor-free margins remains the gold standard management, but it necessitates long-term follow-up because of the high relapse rate (between 36 and 72%) [32]. Imaging is important to establish the effective extension of the tumor, tailor surgery, and monitor relapse. MRI with contrast enhancement by gadolinium is an optimum radiological possibility for diagnosis, especially on T2-weighted images, since it show the lesion as hyperintense relative to muscle [33]. On the other hand, ultrasound can be useful in follow-up monitoring as well. In conclusion, AMM should be evaluated as a differential diagnosis whenever a patient presents with a soft tissue tumor in the vulva–vaginal region, perineum, or pelvis since timely diagnosis with following surgical management (wide surgical excision possibly with free margins) are crucial for the prognosis of these women.

## 9. Conclusions

Soft tissue myxoid tumors constitute a diverse array of lesions characterized by varying degrees of extracellular myxoid matrix. This category of tumors displays considerable variability in their biological behavior and includes benign lesions, tumors prone to recurrence but lacking metastatic potential, and outright malignant neoplasms. Clinically, most lesions emerge as slowly growing polypoid cutaneous lesions and often have to be differentiated from angiomyoblastoma, myxoid neurofibroma, myxoma, spindle cell lipoma, myxoid liposarcoma, leiomyosarcoma and botryoid rhabdo-myosarcoma. Aggressive angiomyxoma typically displace neighboring organs without direct invasion, yet their locally infiltrative nature can eventually result in the invasion of adjacent organs over time, culminating in the development of a large tumor that occupies the abdominal and pelvic cavities. Evidence suggests that AAM affects almost exclusively the genital, perineal, and pelvic regions in women of reproductive age, implicating especially the vulva. Imaging is important to establish the effective extension of the tumor, tailor surgery, and monitor relapse. MRI with contrast enhancement by gadolinium is an optimum radiological possibility for diagnosis, especially on T2-weighted images, since it shows the lesion as hyperintense relative to muscle. Large surgical excision with tumor-free margins remains the gold standard management, but it necessitates long-term follow-up because of the high relapse rate. In conclusion, AAM should be evaluated as a differential diagnosis whenever a patient presents with a soft tissue tumor in the vulva–vaginal region, perineum, or pelvis since timely diagnosis with following surgical management are crucial for the prognosis of these women.

## Figures and Tables

**Figure 1 cancers-16-01375-f001:**
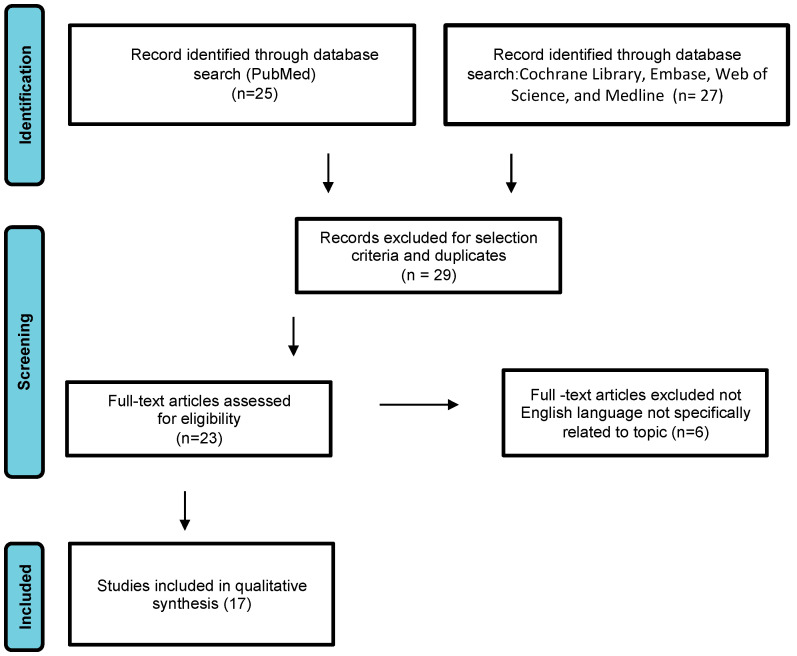
Study flow diagram: PRISMA flow diagram of identification, screening, and inclusion of articles. Systematic literature reviews were selected with standard methods to be briefly presented in the article.

**Figure 2 cancers-16-01375-f002:**
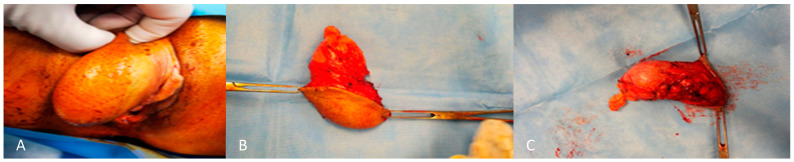
External (**A**) and internal (**B**,**C**) macroscopic aspect of angiomyxoma of the right labium.

**Figure 3 cancers-16-01375-f003:**
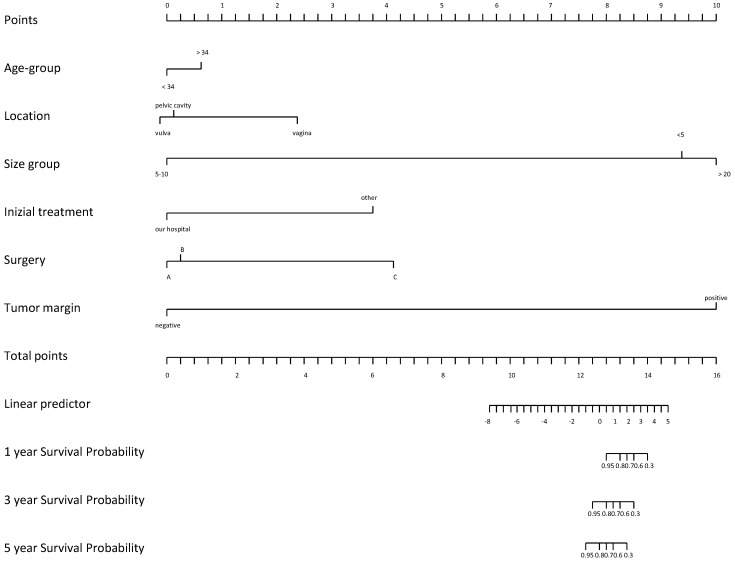
Nomogram on DFS according to initial treatment site (A), surgery procedure (B) and tumor location (C).

**Table 1 cancers-16-01375-t001:** Overview of cases of aggressive angiomyxoma of the lower female genital tract: literature reviewed until February 2023.

References	Sample Size n° / Age	Symptoms / Clinical Aspects	Dimensions/ Site	Primary (P) / Relapse (R)	Treatment	Margins	Histological Features	Immuno-Histochemical Analysis	Follow-Up/ Treatment
Salman MC, 2009 [23]	1 28 yrs	Non-tender mass	3.5 cm × 3 cm RLM	R 8 yrs later	Surgical excision	+	Myxoid stroma/ spindle-shaped cells/ arterioles with thick walls	N.A.	Alive-NED
McCluggage WG, 2006 [24]	1 35 yrs	Menorrhagia, swelling, fluctuant mass	8 cm × 6 cm × 6 cm vulva right side, RLM, centre of the pelvis	P	Surgical excision with TAH+MSO+ GnRHa	+	Myxoid stroma/ spindle-shaped cells/ thick-thin walled blood vessels	ER+ PR−	Alive-NED/GnRHa
Sun NX, 2010 [18]	1 31 yrs	Soft mass	3.2 cm × 1.8 cm RLM (2 cm anterior to the anus)	P	Surgical excision + GnRHa	-	Myxoid stroma/ spindle-shaped cells/ thick-thin walled blood vessels	Vimentin+ SMA + Desmin− S100− ER+ PR+	Alive-NED/GnRHa
Yuan R, 2017 [25]	5 32 yrs (median age 20–52) 1 2 3 4 5	Cystic mass Low abdominal pain Vaginal prolapse Vaginal bleeding after sexual Vaginal mass	3 solid masses (75%) and 1 cystic mass (25%) 18 cm × 10 cm × 9 cm pelvis 23 cm × 20 cm × 10 cm abdomen-pelvis Vagina Vagina Vagina	R (20 months after) P P P P	Surgical excision Surgical excision Surgical excision Surgical excision Surgical excision	+ 2/5 - 3/5	Myxoid stroma/spindle cells-stellate cells/ thick-thin walled blood vessels	SMA + (60%) S100−	Alive (the patients were followed up for 6 to 43 months): 1 RD
Choi H, 2015 [22]	3 49 yrs 31 yrs 36 yrs	Soft mass Mixed echogenic mass, abdominal distension and lower abdominal swelling Soft mass	27 cm LLM 18 cm × 15 cm × 8 cm Pelvis 15 cm × 10 cm × 6 cm left buttock and left perineum, extending to the left retroperitoneum	P P P	Surgical excision Surgical excision Surgical excision	- - -	N.A. Capillaries and cavernous vascular spaces filled with blood and stellate spindle cell proliferation in interstitial tissue Capillaries and cavernous vascular spaces filled with blood and stellate spindle cell proliferation in the interstitial tissue	ER+ Desmin+ CD24+ S100+ CD10+ SMA+ Desmin+ CD34+ CD10+ SMA+ Desmin+ CD34+	Alive-NED Alive-NED Alive-NED
Schwartz PE, 2014 [26]	1 32 yrs	Soft mass	4 cm × 2.5 cm LLM- pelvis- presacral region/ left sciatic nerve	P/R 7 recurrences during 16 yrs, latest in November 2006	Surgical excision + GnRHa	+	Myxoid stroma/ spindle-shape cells/ thick-thin walled blood vessels	ER+ PR+	Alive-NED with GnRHa (free of disease for more than 2 years after discontinuing the leuprolide acetate)
Dahiya K, 2010 [27]	3 27 yrs 30 yrs 32 yrs	Non-tender cyst discharge, dispareunia Tumor mass Cystic mass, swelling and discomfort	5 cm × 4 cm/ right posterior-lateral wall of vagina 10 × 12 × 8 cm/ vulva and paravaginal tissue 3 cm × 4 cm/ vulva	P P P	Surgical excision Surgical excision Surgical excision	- + -	Myxoid tissue & jelly-like material/vascular structure Myxoid-collagenous stroma/thick-walled vessels/spindle-stellate shaped cells Myxoid stroma/ spindle-shaped cells/ thick-thin walled blood vessels	NA NA ND	Alive-NED Lost to F.U. Alive-NED
Raptin C, 2019 [28]	1 24 yrs	Rectocele, perineal swelling	7 cm/ mass of the rectovaginal septum	P	Surgical excision	-	Myxoid stroma/ spindle-shaped cells/ thick-thin walled blood vessels	ER+ PR+ Desmin+ SMA+ CD34+ S100−	R.D. at 1 year: second surgery
Faraj W, 2016 [20]	1 36 yrs	Non-tender mass Pelvic heaviness, urinary urgency	20 cm/ lower back- perineal right area/ right ischiorectal fossa and inferior right perineum/ lung metastasis	R 1 yrs later	Surgical excision	-	Fibro-collagenous and myxoid stroma/ spindled and stellate bland cells	Desmin+ SMA+ S100−	N.A.
Blandamura S, 2003 [29]	1 27 yrs	Mass	20 cm vulva/ lung metastasis	P	Surgical excision	NA	Myxoid stroma spindle and stellate cells/ some vessels	Desmin+ SMA+ S100− Mib1− p53− PR+ ER+ (lung) ER− (vulva)	R.D.-DEAD
Zamani M, 2021 [30]	1 28 yrs	Pedunculated polypoidal mass, soft, spongy painless,	20 × 15 × 10 cm supra and pre pubic/labium majora and minora/invasive lesions in anal canal	R 2 yrs after the first recurrence	Surgical excision/ Decapeptide	-	Stroma without atypia/obvious mitotic activity/variably sized vessels	ER+ PR+	Alive-NED
Wiser A, 2006 [31]	3 25 yrs 40 yrs 48 yrs	Soft mass/ swelling/ Mass Asymptomatic- pedunculated mass	7.6 cm LLM 10 cm right buttock/perineal and intrapelvic 2 cm posterior vaginal fornix	P R P	Surgical excision Surgical excision Surgical excision	- - -	Spindle cells/lack of mitotic activity/ interdigitation protrusion myxoid matrix	NA NA NA	RD: surgical excision/ Alive-NED Alive-NED Alive-NED
Shinohara N, 2004 [32]	1 34 yrs	Gelatinous soft mass	50 × 49 × 19 cm perineum/ pelvic cavity	R 4 yrs later	Surgical excision/GnRHa	+	Myxoid stroma/ spindle-stellate tumor cells/blood vessels	ER+ PR+	R.D.: GnRHa/Alive-NED
Bhandari RN, 2006 [33]	1 74 yrs	Soft, nontender, well mass	8 × 10 cm left gluteus/pelvic floor to within 0.4 cm of the rectum	P	BPS/ RT	N.A.	N.A.	Desmin+ Cytokeratin+ S100− CD34−	Alive-NED 1 year after
Steeper TA, 1983 [1]	9 21 yrs 33 25 26 38 34 32 28 32	Tumor mass Soft mass Lobulated mass NA Painful/“hernia-like” mass Soft mass Swelling/ Soft mass Irregularly rounded mass Soft mass	14 cm × 10 cm × 8 cm vulva 11 cm × 6 cm pelvis/perineum 18 cm × 12 cm vulva/ischiorectal fossa 14 cm × 9 cm × 5 cm vulva 5 cm × 5 cm × 3 cm vulva 60 cm × 20 cm pelvis/gluteal region/retroperitoneum 25 cm × 6 cm vulva/obturator fossa 3 cm × 3 cm vulva 10 cm × 7 cm × 3 cm pelvis/perineum	P R 14–15 yrs later R 3 yrs later P P P R 1 yrs later P P	Surgical excision Surgical excision Surgical excision Surgical excision Surgical excision Incomplete surgical excision Surgical excision Surgical excision Surgical excision	N.A.	Partially or completely encapsulated mass/finger-like tumor projections/ lobulated appearance/ spindled cells in stellate configuration/myxoid background/ prominent vascular pattern	N.A.	R.D. 2 yrs later: surgical excision/ lost to FU Alive-NED Alive-NED Alive-NED Local recurrence-21 months later: Surgical excision-Alive-NED RD 2 months later: surgical excision-NED Alive-NED Alive-NED N.A.
Bégin LR, 1985 [34]	7 63 yrs 21 yrs 36 yrs 47 yrs 30 yrs 36 yrs 32 yrs	Cyst-like lesion Swelling after delivery N.A. Swelling Incidental mass at episiotomy Mass Polypoid lesion	4.3 × 2 × 1.5 cm vagina 14 × 9 × 5 cm vagina/pelvic floor 10 cm vagina 3 × 2 × 1 cm pelvic floor/ vagina/LLM 5.5 cm rectum/LLM perineum/left ischiorectal space 8 × 5 × 5 cm vulva 5 × 3 × 1.5 cm vulva	P P P P P P P	Surgical excision Surgical excision Surgical excision Surgical excision Surgical excision Surgical excision Surgical excision	4/7 +	Myxoid stroma/stellate and spindled-shaped cells/ vascular channels of small or medium-sized arteries and veins.	Actin + CEA− Keratin− FVIII− Prot.S100−	R.D. 16 months later: surgical excision N.A. R.D. (interval NA) 2 R.D. (48,144 months later)-DEAD R.D. 84 months after N.A. R.D. 24 months
Fetsch JF, 1996 [35]	29 (16–70 yrs, median age 34)	Mass/ pain/pressure and pulsating sensations/dyspareunia/increased mass effect with heavy lifting or during menstruation/ pelvic and vague lower abdominal pain/sensation of fullness in the vulva/ urinary frequency	210 cm pelvis/perirectal region/perineum/vulva/buttock/ Bartholin gland region/ retroperitoneum/ inguinal region/LM	P	Surgical excision/ TAH/BSO/ RT	N.A.	Mesenchymal cells/ matrix with collagen/scattered vessels of varying caliber/arborizing vascular pattern absent	Desmin+ (22 pt/22) SMA+ (19 pt/20) MSA+ (16 pt/19) Vimentin+ (17 pt/17) CD34/QBEND-10 + (8 pt/16) ER+ (13 pt/14) PR+ (9 pt/10) S100− (0 pt/20) Ki67 < 1%	8 R.D. 21 NED
Granter SR, 1997 [36]	16 (19–53 yrs, median age 39.5)	Soft mass/pain/Bartholin cyst/inguinal hernia/ abscess	5–23 cm (range) perineal mass/anus/vagina/bladder/rectum/mons pubis/inguinal region/para-urethral	N.A.	Surgical excision	14/16 + 2/16 −	Myxoid stroma/spindled cells/ variable sized vessels	Desmin+ (13/14) SMA+ (10/11) S100−	6 NED 3 N.A. 4 R.D. 1 Alive with symptoms 2 N.A.
York D, 2022 [37]	1 31 yrs	Vaginal itching/pelvic pressure/discomfort	3.5 × 2 × 1.5 cm para-urethral/anterior vaginal wall mass	P	Surgical excision	N.A.	Myxoid stroma/abundant vasculature/ focally infiltrating fibroadipose tissue	N.A.	N.A.
Siassi RM, 1999 [14]	1 63 yrs	Abdominal discomfort/	up to 9.5 by 7 cm pelvis/chest (metastasis)	P	Surgical excision	-	Stellate and spindled cells/ myxo-collagenous matrix	CD34+ ER+ PR+ S100− SMA+ Desmin+ Vimentin+	DEAD
Han-Geurts IJ, 2006 [38]	7 (23–39 yrs, median age 32)	Swelling	left gluteal area/ pelvis/vagina/ LLM/RLM/pubic bone/ischiorectal fossa/perineal area	4P 3R	Surgical excision/embolization/RT	5 − 2 +	Stellate and spindled cells/ myxo-collagenous matrix	ER+ PR+	6 NED 1 RD: NED after treatment
Foust-Wright C, 2012 [39]	1 19 yrs	Asymmetric soft tissue mass/pelvic pain-vaginal protrusion-mild dysuria- occasional urinary stress incontinence	5. × 2.5 × 1 cm/periurethral	P	Surgical excision	-	Spindle-stellate cells/vessels and occasional thick-walled blood vessels	ND	Alive-NED
Srivastava V, 2021 [40]	1 37 yrs	Swelling	10 × 15 cm right vulvar region/perineum/ right ischiorectal fossa	R 3 yrs after	Surgical excision/ Tamoxifen	-	Spindle- stellate cells/myxoid/thin to medium caliber blood vessel/ no atypical mitosis	ER+ PR+ t12q13–15 leading to HMGA2	Alive-NED
Wahid A, 2020 [41]	1 17 yrs	Pain, burning, numbness	7 × 4.5 × 2 cm/ right ischiorectal fossa	P	Surgical excision	-	Spindle shaped cells/myxoid background/ vessels with adventitial thickening and perivascular lymphocytic cuffing with presence of smooth muscle cells	Desmin+ CD34+ S100− SMA−	Alive-NED
Ayati E, 2022 [42]	1 31 yrs	Swelling	13.7 × 6 × 19 cm RLM/ right lateral wall of the vagina and the anal canal	P	Surgical excision	-	Spindle cell/variable-sized blood vessels/ myxoid areas	N.A.	NED
Padmavathy L, 2014 [43]	1 55 yrs	8 cm peduncolate tumor/ 0.5 × 0.5 cm peduncolate tumor	Anterior vaginal wall/ posterior vaginal wall	P	Surgical excision	-	Fibromyxoid stroma/proliferating vascular channels of varying size ulcerated stratified squamous epithelium	Desmin+ CD34+ S100−	N.A.
Li JunHu, 2022 [44]	14 34 yrs (median age 19–58)	11 painless masses (78.6%)/ 2 abdominal pain (14.3%)/ 1 dysuria (7.1%)	5 pelvis (35.7%)/ 5 vulva (35.7%)/ 4 vagina (28.6%)	N.A.	Surgical excision	8− (51.7%) 6+ (42.9%)	N.A.	Desmin+ SMA+ Vimentin+ CD34+ ER+ PR+ S100− (+1/12)	7 RD: 2° surgery+ADJ TRP(GnRHa/CHT/AE)-Alive-NED 7 Alive-NED
Siddiqui SF, 2020 [45]	1 45 yrs	Large mass	13 × 12 × 10 cm vagina/ abdomen/pelvis/ chest (metastasis)	P	BPS/ neoadjuvant GnRH therapy	N.A.	N.A.	Desmin+ CD34+ S100− SMA−	DEAD after therapy

MRV = modified radical vulvectomy; LNM = lymph node metastasis, LND = lymph node dissection; NED = no evidence of disease; NA = not available, PFS = progression-free survival, DEAD = died of disease, R = relapse, P = primary UNK = unknown, LB = labia majora, RLM = right labium majus, LLM = left labium majus, RD = relapse disease.

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
