# Peer review of "Aggressive Angiomyxoma of the Lower Female Genital Tract: A Review of the MITO Rare Tumors Group"

_cancers, 2024, doi:10.3390/cancers16071375_

Round 1

Reviewer 1 Report

Comments and Suggestions for Authors

Dear authors, thank for your work.

This is an interesting paper on an ultra rare and almost anecdotal vulvar tumor. However some improvements should be done before considering it for publication.

line 25: correct typo error

line 42 correct typo error

In introduction paragraph I would add some lines about prognosis difference from the commonest vulvar squamous cell carcinoma (e.g.: 10.1038/s41598-021-85030-x) and why t is important to understand the non aggressive nature of AAM despite its name

Materials and methods: why did you exclude pregnant patients?

line 191 correct typo error

Figure 3 please remove errors and improve readability

Please improve readability of figure 1. I would also suggest adding a column for survival analysis and potential factors related to DFS. I would order papers by cohort numbers or by alphabetical order. Also check for abbreviations.

Line 227 correct typo error

Line 234 correct typo error

Please format the paper with a “conclusion” paragraph. Also I suggest moving paragraph 2-3-4-5 after materials and methods and discussion, in order to provide clearer paper structure

Please remove ref 44 since it is out of context 

Comments on the Quality of English Language

Minor to moderate

Author Response

According to the Reviewer #1’s suggestions:

This is an interesting paper on an ultra rare and almost anecdotal vulvar tumor. However some improvements should be done before considering it for publication.

line 25: correct typo error

Ok done.

line 42 correct typo error

Ok done.

In introduction paragraph I would add some lines about prognosis difference from the commonest vulvar squamous cell carcinoma (e.g.: 10.1038/s41598-021-85030-x) and why t is important to understand the non aggressive nature of AAM despite its name

Ok done.

Materials and methods: why did you exclude pregnant patients?

Because with the national MITO research group we decided in advance the inclusion criteria and to perform two diversified reviews outside of pregnancy and in pregnancy (recently published doi: 10.3390/cancers15133403).

line 191 correct typo error

Ok done.

Figure 3 please remove errors and improve readability

Ok done.

Please improve readability of figure 1. I would also suggest adding a column for survival analysis and potential factors related to DFS. I would order papers by cohort numbers or by alphabetical order. Also check for abbreviations.

Ok, done

Line 227 correct typo error

Ok done.

Line 234 correct typo error

Ok done.

Please format the paper with a “conclusion” paragraph. Also, I suggest moving paragraph 2-3-4-5 after materials and methods and discussion, in order to provide clearer paper structure

Ok done.

Please remove ref 44 since it is out of context

Ok done.

Reviewer 2 Report

Comments and Suggestions for Authors

This manuscript is well-written and may be worthy of publication after the revision.

Comments on the Quality of English Language

This is a comprehensive review of aggressive angiomyxoma of the lower female genital tract. This manuscript has been well written. Please address the following comments to improve the manuscript.

Please check the entire manuscript for more details. There are some typos in the main text (line 25, tumor or tumour, etc.).

Please do not use abbreviations without definition (line 51: AA; line 169: VAA; line 186: HR). Once an abbreviation has been defined in the main text, it can only be used throughout the manuscript.

Lines 40-69, 106-155

This is an excessively lengthy run-on paragraph. The reviewer believes that a paragraph in academic writing is approximately eight to ten lines.

Line 200

What is DAM?

It is unclear why we performed a systematic review of DAM.

Is this systematic review registered with PROSPERO?

Did the authors use the MeSH keywords?

Table 1

Please do not include the metadata in the manuscript. The current version of the table is very difficult to see and the authors need to revise it. The authors need to make an effort to improve the visuality of table.

Author Response

According to the Reviewer #2’s suggestions:

This manuscript is well-written and may be worthy of publication after the revision. This is a comprehensive review of aggressive angiomyxoma of the lower female genital tract. This manuscript has been well written.

Please address the following comments to improve the manuscript.

Thank you for revision.

Please check the entire manuscript for more details. There are some typos in the main text (line 25, tumor or tumour, etc.).

Ok, done.

Please do not use abbreviations without definition (line 51: AA; line 169: VAA; line 186: HR). Once an abbreviation has been defined in the main text, it can only be used throughout the manuscript.

Ok, done.

Lines 40-69, 106-155: This is an excessively lengthy run-on paragraph. The reviewer believes that a paragraph in academic writing is approximately eight to ten lines.

Ok, done.

Line 200: What is DAM?

Sorry, it is typo error that we have corrected.

It is unclear why we performed a systematic review of DAM.

Sorry, it is typo error that we have corrected

Is this systematic review registered with PROSPERO?

We have started the registration on PROSPERO and are waiting for a response.

Table 1 Please do not include the metadata in the manuscript. The current version of the table is very difficult to see and the authors need to revise it. The authors need to make an effort to improve the visuality of table.

We have tried to streamline the table without compromising the numerous data collected from the literature.

We hope that this new version can be more appreciated.

Reviewer 3 Report

Comments and Suggestions for Authors

The article is comprehensive and well-written.  There are two problems noted:

1- The formatting of the charts should be rectified to have the space better utilized.

2- Page 14 Line 227, the word 'Array' is misspelled

Author Response

According to the Reviewer #3’s suggestions:

The article is comprehensive and well-written. There are two problems noted:
1- The formatting of the charts should be rectified to have the space better utilized; 2- Page 14 Line 227, the word 'Array' is misspelled

Ok, done. Thank you for revision.

Round 2

Reviewer 1 Report

Comments and Suggestions for Authors

Revision points highlighted by reviewers has been addressed. Now the paper deserves publication

Comments on the Quality of English Language

Minor

Reviewer 2 Report

Comments and Suggestions for Authors

The authors have revised the manuscript according to the reviewers' suggestions. Nevertheless, please do not register PROSPERO after starting the systematic literature review. In this case, please clarify that this systematic review was not registered.

Comments on the Quality of English Language

Acceptable.